

# Short communication: Potential of Sentinel 1 InSAR and offset tracking in monitoring post-cyclonic landslides activities in Reunion Island.

Marcello de Michele[1], Daniel Raucoules[1], Claire Rault[2], Bertrand Aunay[2] and Michael Foumelis[1,3]

[1]BRGM, Geophysical Imagery and Remote Sensing Unit, Orleans, 45000, France.

[2]BRGM, Direction de Actions Territoriales, Saint Denis, La Réunion, 97400, France.

[3]Aristotle University of Thessaloniki, Department of Physical and Environmental Geography, 541 24 Thessaloniki, Greece.

*Correspondence to*: Marcello de Michele (m.demichele@brgm.fr)

**Abstract**

This study examined the results of an interferometric Synthetic Aperture Radar (InSAR) and SAR Offset Tracking (OT) study on Cirque de Salazie (CdS), Reunion Island, France, within the context of the RENOVRISK project, a multidisciplinary programme to study the cyclonic risks in the South-West Indian Ocean. CdS is one of the denser populated areas in Reunion Island. One of the aims of the project was to assess whether Sentinel 1 SAR methods could be used to measure landslide motion and/or accelerations due to post cyclonic activity on CdS. We concentrated on the post 2017 cyclonic event. We used the Copernicus Sentinel 1 data, acquired between 30/10/2017 and 06/11 2018. Sentinel 1 is a C-band SAR, and its signal can be severely affected by the presence of changing vegetation between two SAR acquisitions, particularly in CdS, where the vegetation canopy is well developed. This is why C-band radars such as the ones onboard Radarsat or Envisat, characterized by low acquisition frequency (24 and 36 days, respectively), could not be routinely used on CdS to measure landslide motion with InSAR in the past. In this study, we used InSAR and OT techniques applied to Sentinel 1 SAR. We find that C-band SAR onboard Sentinel 1 can be used to monitor landslide motion in densely vegetated areas, thanks to its high acquisition frequency (12 days). OT stacking reveals a useful complement to InSAR, especially in mapping fast moving areas. In particular, we can highlight ground motion in the Hell-Bourg, Ile à Vidot, Grand Ilet, Camp Pierrot, and Bellier landslides.

## 1. Introduction and study area

Landslide and erosion processes are causes of major concern to population and infrastructures on Reunion Island. These processes are led by the tropical climate of the island. The hydrological regime of the rivers is distinct owing to the coexistence of several major parameters that predispose it to extreme vulnerability. Holding almost all the world records for rainfall between 12 h (1170 mm) and 15 days (6083 mm), the island has a marked relief with a peak at 3,069 m, with exceptional cliffs that reach 1500 m in height.



CdS is the rainiest of the large erosional depressions on Reunion Island (Pohl et al., 2016) with an average annual
cumulative rainfall of approximately 3,100 mm since 1963; a minimum of 698 mm was recorded in 1990, and a
maximum of 5,893 mm was recorded in 1980.
This depression is surrounded by steep rock cliffs and filled with epiclastic material. Intense river erosion incises
deep valleys and has produced several isolated plateaus across the cirque.
The morphology, geology, and climate make CdS prone to erosion and ground movements. At least 19% of its
slope has been affected by landslides (Rault et al., 2022). The active landslides range from a large slow-moving
landslide of hundred million cubic meters to rapid and catastrophic slope failure with a volume exceeding one
million cubic meters.
Eleven slow moving landslides are identified in CdS (Figure 1). Their displacement rates range from a few cm/yr
to 1.15 m/yr and can accelerate after intense rainfall events, particularly because of cyclonic activity (e.g Belle et
al., 2014). These landslides are commonly observed on plateaus. They cover areas that vary from tens of thousands
of square m to several square km. Hell-Bourg (HB) and Grand-Ilet (GI) are the largest inhabited slow-moving
landslides in the cirque with volumes of 225 $\times 10^6$ m$^3$ and 215$\times 10^6$ m$^3$, respectively (e.g. Rault et al, 2022).

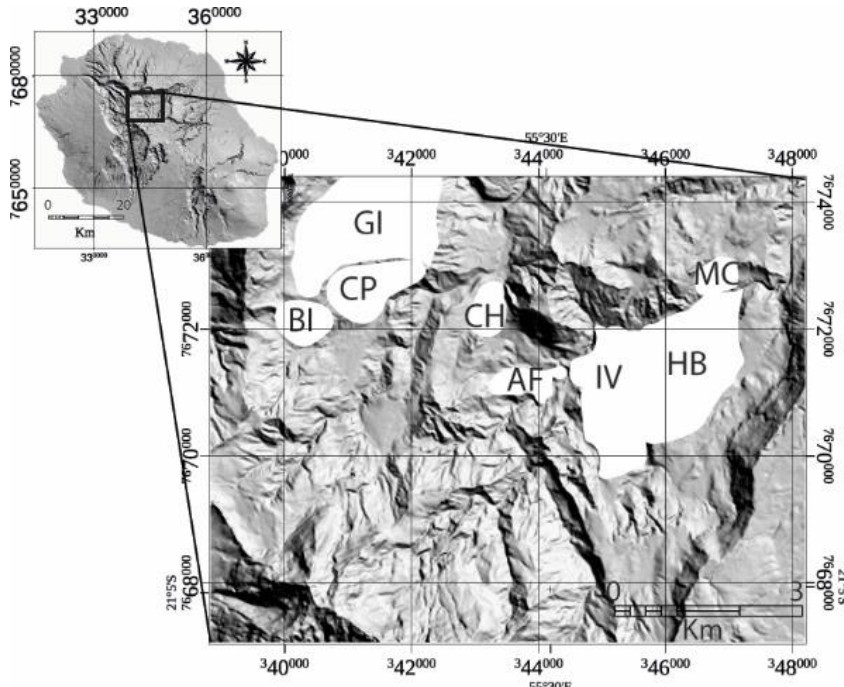


**Figure 1. Map of the study area. Slow moving landslides in the area, from existing catalogues, are highlighted in white.**
**MC : Mare à citron, HB:  Hell-Bourg,  IV: Ilet à Vidot, AF: Affouche, CH: Chemin Henry, BL: Bellier, CP: Camp**
**Pierrot, and GI: Grand-Ilet.**



Slow-moving landslides can trigger secondary landslides along their steep scarps that eventually border rivers,
leading to increased solid loads. Therefore, slow-moving landslides are not only responsible for significant damage
to houses and infrastructure but are also involved in the formation of torrential flows and river dams (Liébault et
al. 2010, Tulet et al., 2021). Thus, understanding their kinematics/dynamics is essential for hazard and risk
mitigation at the scale of Reunion Island.
SAR methods demonstrated useful in highlighting landslides kinematics from space (e.g. Aslan et al., 2020 –and
references therein).  While C band SAR can be used to comprehensively measure ground displacement, the study
area is particularly challenging for C band SAR since it is densely vegetated. In this study, we test whether InSAR
and OT techniques can be used to measure landslide kinematics in densely vegetated areas. Both techniques require
temporal signal coherence and therefore are usually not adapted densely in vegetated areas. Particularly, C-band
SAR signal can be severely affected by the presence of changing vegetation between two SAR acquisitions,
particularly in CdS, where the vegetation canopy is well developed. This is why C-band radars such as the ones
onboard Radarsat or Envisat, characterized by low acquisition frequency (24 and 36 days, respectively), could not
be routinely used on CdS to measure landslide motion with InSAR in the past. This limitation can be palliated
under certain circumstances, for instance improving repetition frequency between two or more satellite
acquisitions. In this study, we exploit the improved repetition frequency of Sentinel-1 SAR to test InSAR and OT
techniques in CdS after a cyclonic event in 2018. If successful, this study can serve as a demonstrator.

**1.1 Past studies regarding ground motion from space in CdS**
Few studies have used spaceborne remote sensing techniques to report on ground instabilities in CdS. Delacourt
*et al.* (2009) used a combination of optical (Spot 5 and aerial imagery) and synthetic aperture radar (SAR) data
(JERS-1 and Radarsat data) to measure ground motion on the Hell Bourg landslide. They applied the two
techniques to assess their performance and quantify ground motion associated with this landslide. They found an
average displacement of approximately 0.5 m/yr from 1997–2002. Le Bivic *et al.* (2017) used two pairs of ortho-
rectified SPOT-5 images at 2.5 m resolution on the Hell Bourg landslide. The first pair of images spanned the
period between 2002 and 2005. The second pair of images spanned the period 2006–2008. They reported that
during 2002–2005, the OT method yielded ground motion within the signal noise; they deduced that landslide
activity was low. From 2006–2008, they measured a maximum displacement of 8.5 ±2 m (possibly due to the
storm Gamede).
Raucoules *et al.* (2016) used high resolution X-band SAR data from the TerraSAR-X satellite from 2010–2011.
They combined ascending and descending OT maps to extract the three dimensional displacement field of the HB
and GI landslides. They reported that ground displacement reached $1 \pm 0.25$ m/y vertically and $0.65 \pm 0.25$ m/y
horizontally. They also used InSAR combined with X-band data from the Cosmo-SKYMED satellite to measure
centimetric displacements on the borders of the HB and GI landslides. The X-band InSAR signal was incoherent
elsewhere.
Raucoules *et al.* (2018; 2020) used space-borne high-resolution L-band SAR (ALOS-2/PALSAR2 data in
StripMap SM1 mode) both with interferometric synthetic aperture radar (InSAR) and OT. They derived two
components of the displacement field for the HB landslide. The displacement reached approximately 1 m/y from



2014–2016. They reported that L-band SAR performed significantly better than the C-band SAR available at the
time of the study.

**1.2 Aim of this study**
Landslides displacement rates in CdS can accelerate after intense rainfall events, particularly during cyclonic
activity. Their kinematic might change during such extreme events; new landslides might appear. Global
Positioning System (GPS) in the study area yields precise time series at the measurement stations. SAR methods
are potentially able to spatialize the ground motion information and might reveal ground motion in unexpected
areas. The aim of this study is to assess whether Sentinel 1 SAR methods – both InSAR and OT- could be used to
measure landslide motion and accelerations caused by post cyclonic activity in a densely vegetated area such as
CdS. This study complements the one by Raucoules *et al.* (2018, 2020), who used L-band InSAR and OT to
measure ground motion in CdS. The measurement of ground motion with C band SAR in densely vegetated areas
is challenging because the radar waves interact with the vegetation canopy and may yield an incoherent InSAR
signal if temporal changes occur between the two SAR scenes. Therefore, InSAR signal coherence largely depends
on the revisit time of the satellite. The shorter the revisit time is, the higher the InSAR signal coherence, and the
faster the ground motion has to be in order to be measured by InSAR. The history of C-band SAR data over la
Réunion island is non-linear. The European Space Agency (ESA) Earth Remote Sensing Satellite (ERS-1and ERS-
2) SAR platforms did not cover La Réunion Island owing to orbit incompatibilities. The Canadian Space Agency
Radarsat 1-2 has flown over la Réunion, but they acquired data every 24 days (Delacourt et al., 2009), which
makes the InSAR signal incoherent in densely vegetated areas such as the CdS. Similarly, the ESA Advance SAR
(ASAR) sensor onboard the ENVISAT satellite acquired data every 35 days. Therefore, the SAR interferometric
signal was incoherent for this C-band radar with a quasi-monthly repeat cycle. The Copernicus Sentinel 1 satellite
can resolve these problems. Sentinel 1 hosts a C-band SAR (wavelength = 5.5 cm) whose interferometric signal is
usually incoherent over densely vegetated areas; however, the high repetition frequency of Sentinel 1 (12 days in
La Réunion, 6 days in mainland Europe until 2021) makes the InSAR signal potentially suitable for measuring
land displacements in densely vegetated areas (e.g. Aslan et al., 2020). It could be complementary or alternative
to L-band SAR interferometry (Delacourt et al., 2009; Raucoules et al. 2020) and *in-situ* techniques in La Réunion.

**1.3 InSAR and Offset Tracking techniques**
In this study, we designed the experiment as follows. We applied two SAR methods, InSAR and OT. These two
methods could be complementary because InSAR can measure slow moving landslides, typically a small fraction
of the employed wavelength, while OT could measure large ground motions, higher than the pixel size of the SAR
scene. In contrast, OT may be limited in the measurement of small ground motions, depending on the pixel size,
because the nominal lower bound precision is $1/10^{th}$ of the pixel size of the image employed on a single
correlogram.
InSAR methods rely on the measurement of the changes in SAR phases among multiple SAR scenes using
interferometric processing (e.g. Massonnet and Feigl, 1998). It is a widely used methodology to measure ground
displacements from space, in many disciplines related to tectonics (e.g. Elliot et al., 2020), volcanology (e.g.



Doubre et al., 2017) and gravitational failures (e.g. Aslan et al., 2020). Here, we used the stacking procedure
implemented in the Gamma processing chain. The stacking procedure combines multiple SAR scenes and yields
ground displacement rate in the form of ground velocity map. It is used to estimate the linear rate of differential
phase starting from a set of unwrapped differential interferograms. The individual interferogram phases are
weighted by the time interval in estimating the phase rate. The underlying assumption is that atmospheric statistics
are stationary for the set of interferograms.
OT is a sub-pixel image correlation technique. This technique matches two or more images at each point on a grid,
analyzing the degrees of local correlation at each step. Differences in the local instantaneous frequency of the
images result in sub-pixel spatial differences in ground patterns. Measurements must be performed with subpixel
accuracy because the amplitude of the ground displacement is often lower than the resolution of the images,
depending on the sensor used.
Sub-pixel image correlation technique for measuring ground surface displacements can be applied to optical (e.g.
Michel et al., 1999; de Michele and Briole, 2007) and SAR amplitude images (e.g. Michel and Rignot, 1999; de
Michele et al., 2010a, 2010b). The main differences between optical and SAR images are caused by the oblique
SAR acquisition geometry. Therefore, instead of having east-west and north-south offsets as in optical OT, SAR
OT has with slant range and azimuth offsets. Slant range is the line of sight (LOS) direction, or look angle of the
satellite. The azimuth is the flying direction of the satellite (nominally 98° for Sentinel 1; nearly north-south).
Moreover, azimuth offsets are «topography free», and slant range offsets are calculated in the LOS and, therefore,
contain a contribution from vertical offsets, depending on the viewing angle. Thus, OT yields two components of
the deformation field for one SAR scene. For details on this methodology, please refer to Michel & Rignot (1999)
and Michel et al., (1999), who used it with shuttle imaging radar (SIR-C) ERS radar amplitude images, and
Raucoules et al. (2013), who applied the OT method to multi temporal SAR images at La Vallette landslide. The
OT technique provides a measurement of the ground displacement from the analysis of the geometrical
deformation between the two SAR amplitude images. Usually SAR images with a small baseline are chosen to
reduce the stereoscopic effect and geometric decorrelation. For Sentinel 1, the orbit tubes are steered within 100
m maximum. Therefore, the topographic contribution to the OT in the LOS direction is negligible. In this study,
we estimated the range and azimuth offset fields using cross correlation optimization of the input intensity images.
This algorithm was implemented in the GAMMA software with the name of "offsets tracking" (e.g. Strozzi et al.,
2002). GAMMA is a standard SAR processing software, which results have been validated among other available
SAR processing chains (e.g. Raucoules et al., 2009).

## 2. Data and processing steps

At the time of this study experimental design, the Sentinel-1 mission comprised a constellation of two polar-
orbiting satellites, operating day and night, hosting a C-band synthetic aperture radar, enabling them to acquire
imagery regardless of the weather. We used 26 Sentinel 1 data in descending, stripmap mode, acquired every 12
days from October 30, 2017, to November 06, 2018. Sentinel 1 data are provided by the European Space Agency,
within the Copernicus Program of the European Union. Technically, we downloaded them as Single Look
Complex (SLC) data from the Copernicus Scientific Data Hub (https://scihub.copernicus.eu). We did not use the



ascending mode, as it is less sensitive to ground motion in CdS given the SAR shadow and the unfavorable relative
geometry between the SAR orbit and ground motion (e.g. Raucoules et al., 2020).
First, we co-registered the SAR data. Then, we created differential interferograms with data pairs spanning 12 days
intervals. The topographic contribution to the interferometric phase was modelled using a DEM from the Shuttle
Radar Topography Mission (SRTM). We unwrapped the differential interferograms with the minimum cost flow
algorithm (Constantini, 1998). Depending on the water vapor content in the air, there may be an interferometric
phase delay due to the SAR signal crossing multiple tropospheric layers. To a first approximation, this delay is
proportional to the topographic slope. Therefore, we corrected the tropospheric contribution to the interferometric
phase by linear regression with the DEM. At this point, we performed the stacking procedure, producing a velocity
map. Then, we orthorectified the velocity map. The results are shown in Figure 2.
For the OT procedure, we began from the co-registered SAR data. Instead of the phases, we used the amplitude of
the SAR signal and extracted multi look complex images (MLI) for each acquisition date. Multi look processing
degrades the image resolution but reduces the image speckle. Because OT sensitivity to ground motion depends
on the pixel resolution, we required a tradeoff between image noise and multi looking. For this study, our choice
was multi looking with a factor of 3 in azimuth and 2 in range directions. This led to pixel sizes of 8.2 m in the
azimuth direction and 7.9 m in the range directions. We used a correlation window of 128X128 pixels and searched
for 1024 measures in range and azimuth respectively. The OT technique is nominally less affected by temporal
signal decorrelation than the InSAR technique. Therefore, we used all possible image couples, leading to the
creation of 351 correlograms in both range and azimuth directions. Then, we applied a stacking procedure to create
one velocity map in the range direction and one in the azimuth direction. Finally, we orthorectified the results.

## 3. Results

We recognized and mapped three landslide areas in Salazie, active during the study period. Hell Bourg, Ilet à Vidot
and an area that was considered stabilized or dormant, possibly corresponding to Crete de Salaze.
Hell Bourg and Ilet a Vidot are two major inhabited compound landslides of Salazie. These landslides move
continuously and typically accelerate following heavy rainfall. They occupy approximately 10% of the CdS surface
area. They all stand on volcanoclastic material interpreted as volcanic debris-avalanche deposits or as debris-
flow/mud-flow deposits by Rault et al. (2022). Hell Bourg is a compound landslide covering a surface of
approximately 2.8 km². Ilet a Vidot is an active plateau located northwest of Hell Bourg and covers an area of 2.3
km².
In the next paragraph we describe both InSAR and OT results in more details.

### 3.1 InSAR

The stacking InSAR results are shown in Figure 2. They are presented as a mean velocity map, where ground
motion is measured in the LOS of the satellite. InSAR is only sensitive to ground motion occurring in the LOS
direction. Motion away from the satellite is represented in blue. Motion towards the satellite is represented in red.
Complex combinations of those two directions of motion may lead to zero apparent motion on the InSAR velocity



map. From Figure 2, we observe manifold improvements with respect to velocity maps produced from former C-
band InSAR missions (Delacourt *et al.*, 2009). First, we noticed that the InSAR signal was coherent over the study
area, which was not expected given the densely vegetated tropical area. Second, we can see a clear pattern of
ground motion in the Hell-Bourg area, approximately ± 0.5 m/yr and spatially consistent with ground observations.
Figure 2 also shows ground motion on the Ilet à Vidot (IV) landslide. The InSAR signal direction on both HB and
IV landslides is quite complex. This suggests the existence of complex internal landslides kinematics such as
stretching of the main landslide body in HB and dismantling of IV plateau on both sides by landslides moving
either eastward or westward, as expected for a compound slide and identified in the field by Rault et al. (2022). It
also suggests a rotational component of ground motion. In this study, the InSAR signal was surprisingly coherent
with a certain level of noise. We calculated the noise as ±2 cm/yr on the InSAR velocity map. To minimize
coherence loss, we used interferograms with 12 days times span only, excluding larger time-span interferograms.
Owing to these limitations, we were unable to capture very slow ground motions (< 2 cm/yr).

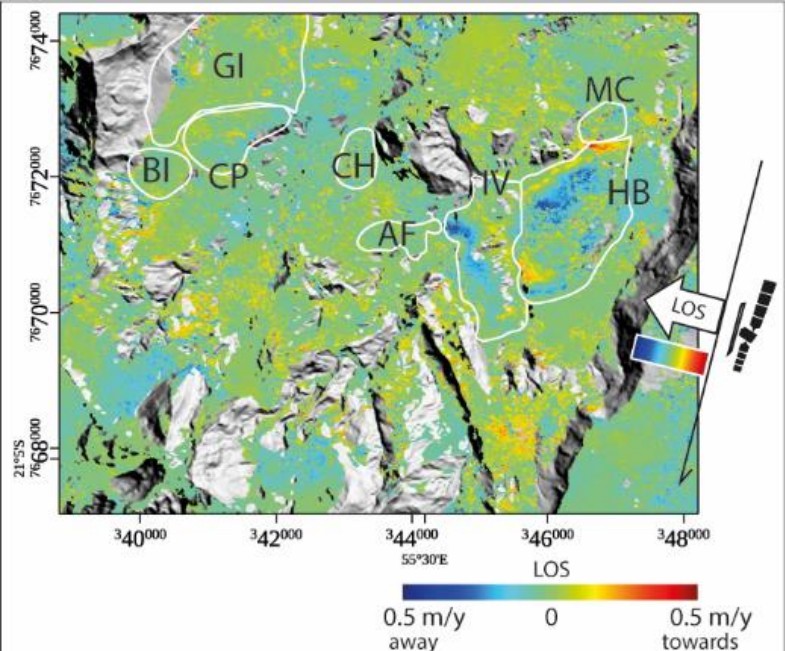


**Figure 2.  InSAR velocity map in the LOS of the satellite, using the descending mode. A ground displacement pattern**
**is clearly visible in the Hell-Bourg area.**

**3.2 Offset tracking Azimuth and Range**
We show the OT results in Figure 3. Because OT is a subpixel correlation technique, its precision depends on the
image pixel size. Nominally, the correlator implemented into the GAMMA processing chain is as precise as $1/10^{th}$
of the pixel size (e.g. Raucoules *et al*., 2019). Therefore, because the MLI pixel sizes were 8.2 m in the azimuth

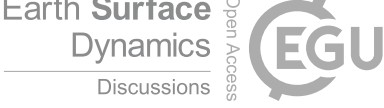

226 direction and 7.9 m in the range directions, we can expect precisions of the orders of 0.8 m in a single correlograms.

227 Thus, we cannot use OT to measure ground motions smaller than 0.8 m on a single correlogram. The precision

228 increases by applying the stacking procedure. Moreover, the stacking procedure can compensate the component

229 of the pixel offsets that may be caused by non-zero baselines between Sentinel 1 orbits, which is proportional to

230 topography. Furthermore, OT does not have higher limits for ground motion detection. This latter characteristic is

231 particularly helpful in CdS, where metric ground motion is expected, particularly within the HB area.

232

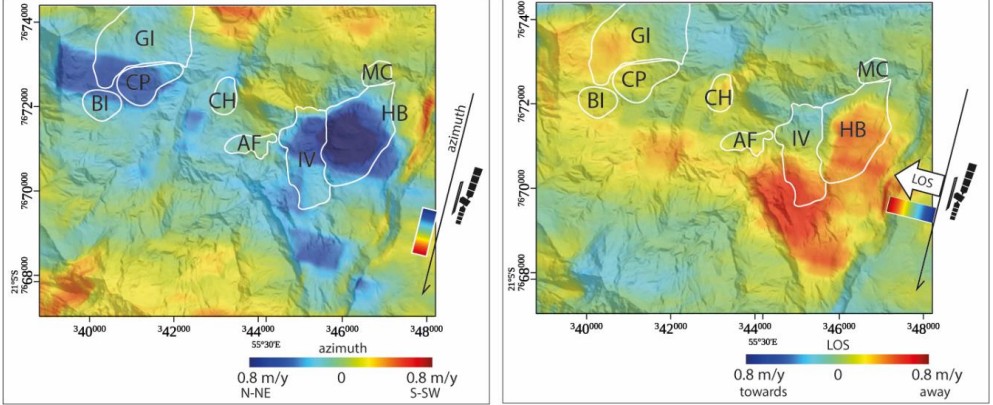

234 *Figure 3. Azimuth (left) and LOS (right) OT velocity maps.*

235 The first observation from Figure 3 is that the OT stacking procedure applied to Sentinel 1 MLIs provides

236 meaningful results, both in the azimuth and slant range directions. From Figure 3, we find that the fastest ground

237 motion in the azimuth direction is localized on the Hell Bourg landslide, N-NW section of the IV landslide, and

238 Camp Pierrot (CP) landslide. Ground motion in Hell Bourg can reach 1 m/yr in the azimuth direction. The OT

239 signal in the azimuth direction is also visible on a central section of the CP and IV landslides, as fast as 0.7 m/yr.

240 In the slant range direction, the OT ground motion signal is localized in the HB landslide, reaching 0.8 m/yr, away

241 from the satellite. Figure 3 highlights motion on the S-SE sections of the GI and IV landslides. We also observe

242 an unexpected pattern of ground motion S-SE of HB and south of IV, that was consistent with the geomorphology

243 of the area but situated in a non-instrumented, uninhabited area on the ground. This signal is intriguing and must

244 be validated against *in situ* observations. It might correspond to an area that was considered stabilized or dormant,

245 called Crete de Salaze.

246 To further investigate this latter signal, we considered the horizontal component of LOS motion only, combined

247 with the azimuth velocities, to extract OT horizontal ground velocities regardless of the motion direction. The

248 hypothesis holds in CdS because the horizontal motion (in HB for instance) is almost 7 times larger than the

249 vertical velocity. Therefore, we applied :

250 $$|v_h| = \sqrt{AZ^2 + (LOS/\sin i)^2}, \tag{1}$$

251 to extract the horizontal velocity map, $|v_h|$, where i is the LOS viewing angle (37° in this case study), AZ is the

252 azimuth offset and LOS is the range offset (Figure 4). Figure 4 shows that the OT detectable ground motion is

253 concentrated in the HB, IV, and south of IV landslides. There is also a weaker but noticeable ground motion signal

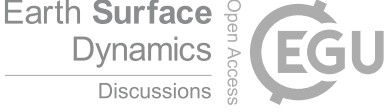

at the GI and CP landslides. Moreover, there is a marked signal S-SE of IV. This area, considered stabilized or
dormant, must be investigated further, the OT signal may be due to a post cyclonic burst of ground motion.

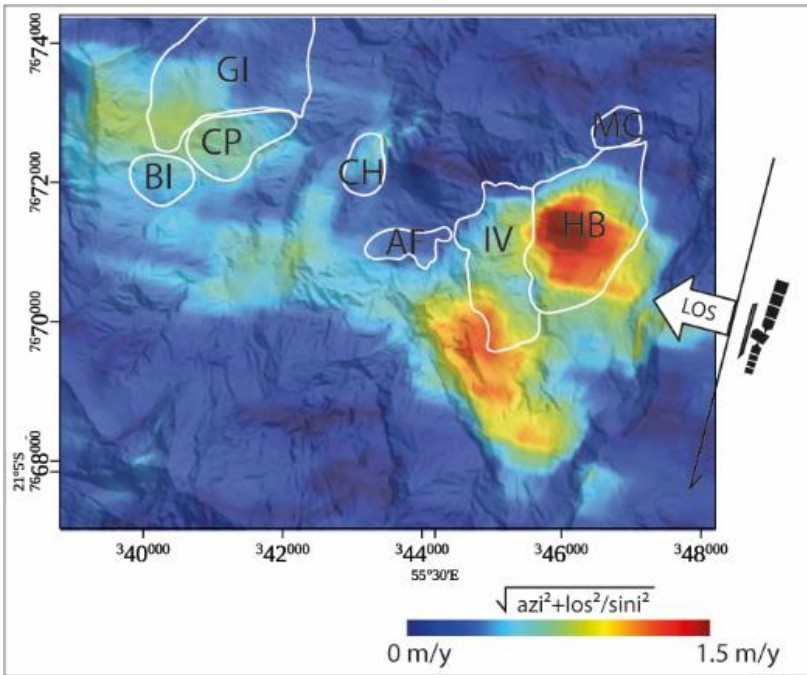


**Figure 4. Amplitude of the OT horizontal ground velocities independent of direction. We consider only the horizontal component of LOS : sqrt(AZ²+LOS²/sini². i is the LOS viewing angle (37°). The hypothesis is that the horizontal motion in CdS (in HB for instance) is almost 7 times larger than the vertical velocity.**


**4. Comparison with global positioning system (GPS) data**

To gain some insight into the accuracy of SAR velocities maps, we performed a cross-comparison with global
positioning system (GPS) campaigns available at CdS. In this exercise, we compared SAR velocities with GPS
velocities acquired over the time span May 2018 - February 2020. The Global Navigation satellite system (GNSS)
velocities were calculated using 93 geodetic markers across the cirque (Figure 5). They were obtained from the
position of the markers measured with a differential GPS during two campaigns of measurements: May 2018 and
January 2020.

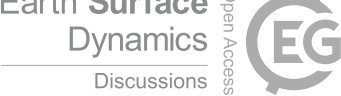

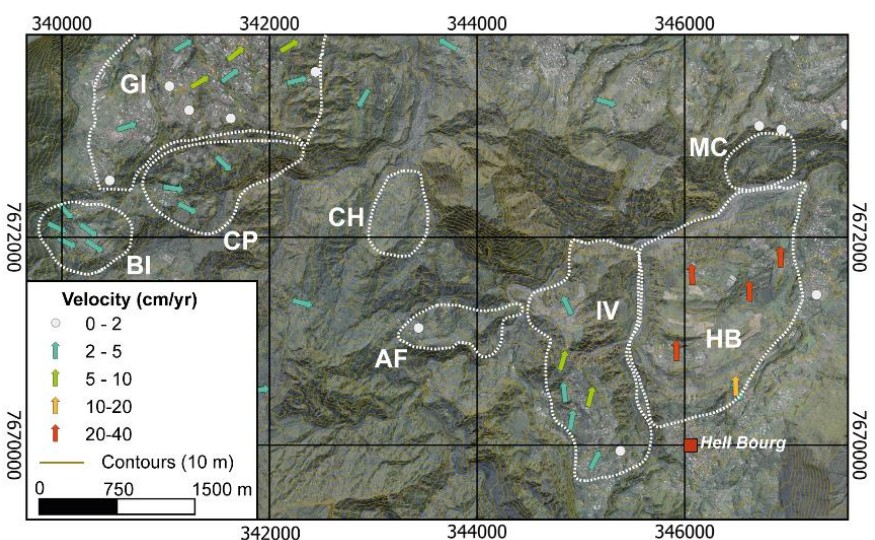


**Figure 5. Locations and horizontal velocities of GPS stations. Velocities refers to campaigns in May 2008 and January**
271                                                    **2020.**


GPS measurement accuracy varies from one site to another depending on environmental factors (e.g., vegetation,
proximity to buildings, cirque cliffs, and steep ramparts). For each campaign of measurement, the position of a
benchmark was measured four successive times. The final position of the benchmark is the average of these four
measurements. Measurements with deviations of more than 5 cm in altimetry and 3 cm in planimetry were removed
from the dataset. The positioning accuracies are of the order of 2 cm in planimetry and less than 5 cm in altimetry
(Mazué et al., 2013).
To compare GPS and SAR velocities (both InSAR and OT), we must project the GPS x, y, and z values into LOS
(= OT range) and OT azimuth direction, by considering that Sentinel 1 had a heading of  -167.66° South with a
look angle of 36.93°. We then obtained GPS$_{sar}$ values in LOS, range and azimuth direction. Then, we discretized
the GPS$_{sar}$ values into a number of intervals of 0.03 m width. For each interval, we calculated the median and
identified the geographic location of each point in the interval. At those points, we extracted median GPS$_{sar}$ and
SAR values. Then, we plotted GPS$_{sar}$ *versus* SAR for each interval. The results are scatterplots showing GPS
(LOS) vs InSAR (Figure 6) and GPS (range; azimuth) vs SAR OT (range, azimuth) in Figure 7.
From Figure 6, we see that the comparison between InSAR velocities and GPS$_{sar}$ velocities is satisfactory.
However, there was a general underestimation of motion for the InSAR technique (i.e., InSAR=0.8xGPS). This
may be due to an uncompensated residual ramp in the InSAR velocity map.



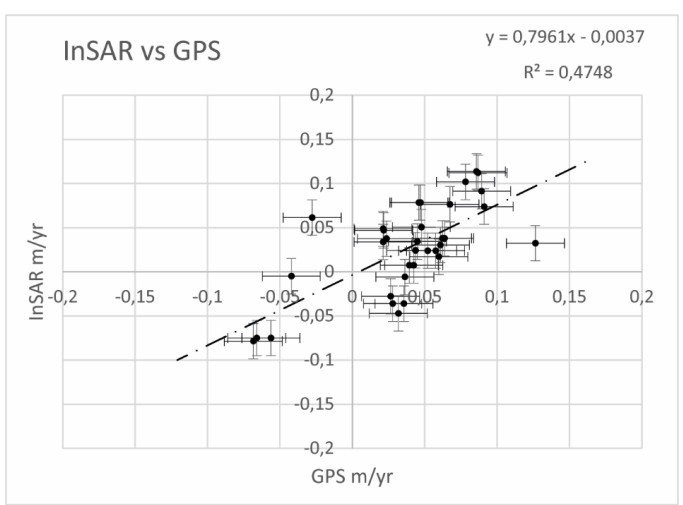


**Figure 6. InSAR VS GPS (LOS) data (m/yr). InSAR is not sensitive to velocities higher than 2 cm/yr. Therefore, we**
**mask all values that are related to GPS values higher than |2| cm per year.**




In Figure 7, we show the comparison between $GPS_{sar}$ and OT velocities in range and azimuth directions. We see


that the comparison between SAR and $GPS_{sar}$ velocities is good. There is a general underestimation of motion by


the OT technique in the azimuth direction (i.e., OTazimuth=0.5xGPS$_{sar}$). This may be due to an uncompensated


residual ramp in the OTs velocity maps.



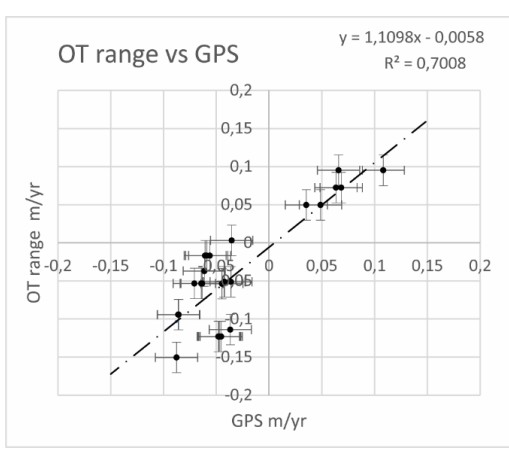 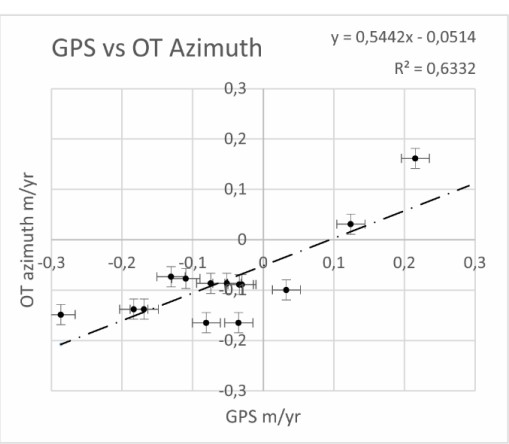



**Figure 7. Left. $GPS_{sar}$ vs OT in the range direction (m/yr). Right: $GPS_{sar}$ vs OT in the azimuth direction (m/yr). Masked**
**GPS values between ±3 cm per year**






### 5. Discussion, conclusions, and perspectives

In this study, we applied Sentinel 1 SAR analysis, both InSAR and OT, to the measurement of ground motion on Cirque de Salazie, Reunion Island, France. Thanks to the high repetition frequency of Sentinel 1, the C-band InSAR signal was coherent in this region, in contrast with past C-band studies on the study area. This result allows us to produce interpretable velocity maps, both with InSAR and OT techniques. The InSAR velocity map provided spatially detailed ground velocities in the LOS of the satellite, with precision reaching a fraction of the SAR wavelength. The comparison between InSAR and GPS velocities is satisfactory. The Sentinel 1 MLIs OT technique provided useful measurements. Nevertheless, the comparison between OT and GPS velocities highlighted several biases that require a more detailed investigation. The biases may be due to a residual orbital ramp in the OT velocities but also the fact that OT precision is a function of the MLI pixel size. Because the InSAR and OT techniques provided "relative" measurements, the biases with respect to GPS suggested that absolute calibration of SAR maps is needed to compare SAR and GPS results.

Within the InSAR and OT detection limits and the period of SAR measurements (October 2017–November 2018) this study displayed the ground motion and internal kinematics on HB landslide, the N-NW section of IV landslide, and CP landslide. Moreover, we point out an unexpected pattern of ground motion S-SE of HB and South of IV, consistent with the geomorphology of the area but situated in a non-instrumented, uninhabited area on the ground. We suggest that it might correspond to an area that was considered stabilized or dormant, called Crete de Salaze. Motion in this area might have been reactivated as a consequence of heavy rainfall and thus represents a post cyclonic burst of ground motion.

Three GPS sites on the South-Western part of IV landslides indicates ground motion in this sector of CdS. Therefore, this noticeable signal requires further consideration and investigation.

We need precise time series of this ground motion, localized on those three specific areas, to discriminate whether we can highlight rheological changes due to the post-cyclonic activity.

There are rooms for methodological improvement:

Even though the Sentinel 1 ascending mode is less adapted in CdS due to shadow and layover effects depending on the look angle, it may be possible to exploit the few coherent pixels to extract the vertical and east-west components of ground displacement, by a combination of Sentinel 1 InSAR in ascending and descending modes. The exploitation of both InSAR and OT from Sentinel 1 and other SAR missions (e. g. ALOS 2, TerraSAR X) would improve data coverage, spatially and temporally. OT with Sentinel 1 data revealed precious to measure large ground motion in CdS. In this study, we calculated OT on MLIs 3-2; we could run an experiment to determine whether OT from MLIs 2-1 or even MLIs 1-1 could lead to more accurate results.

In conclusion, Sentinel 1 InSAR and OT present high potential for routine monitoring on CdS, as a complement to *in situ* techniques (i e., Rault et al., 2022). This study presented a solid premise for the future exploitation of the European Ground Motion Service (EGMS) of the European Union Copernicus program, based on Sentinel 1 InSAR, in CdS.



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
