# Peer review of "tracking in monitoring post-cyclonic landslides activities in"

_Earth Surface Dynamics, 2022_

## Author Response (AR1)

**We are thankful to the Reviewers for the time they spent to evaluate our work. We do appreciate their positive suggestions to improve our work.**

**We took into account all the comments and we improved the manuscript accordingly. We also performed miscellaneous editing in the new version of the manuscript. Here, we reply to the all the suggestions for improvement made by Reviewer 2. Our reply are in bold typography.**

Reviewer1.
The present short communication deal with the combine use interferometric Synthetic Aperture Radar (InSAR) and Offset Tracking (OT) processing techniques od SAR images to investigate the landslide motion and/or accelerations due to post cyclonic activity. The analysis was performed within a multidisciplinary research project (called RENOVRISK) over a vegetated area of the Cirque de Salazie (Reunion Island, France) using Sentinel-1 SAR data. The main novelty of the study is the jointly use of both InSAR and OT processing method that applied to C-band SAR images with high acquisition frequency (12 days for Sentinel constellation), can be used to monitor slow-moving landslide motion also densely vegetated areas (generally bed covered) as well to identify and mapping fast moving areas. The paper is interesting for people involved in the mapping of new moving areas and in the kinematic characterization of mapped landslide areas. The designed research fits well with the purpose of the Journal with results that are encouraging which show a promising use of the combined proposed approach to detect moving area in high vegetated condition. From my point of view the paper can be considered for the publication in the present form.

**Authors: We thank Reviewer #1 for his very positive opinion of our work.**

Reviewer 2.
The letter proposes the use of InSAR and offset tracking techniques on Sentinel-1 SAR data to monitor landslides in a vegetated area (Reunion Island). The work is interesting and suitable for publication in this journal, in the following detailed comments are provided.

**Authors: we are thankful to the reviewer for the positive comments and the suggestions for improving the manuscript.**

Reviewer 2.
As a general point, the quality of the figures should be improved, as they appear blurred.
**Authors: this might have been a visualization issue. We have tried to improve the image resolutions as suggested by the reviewer –even though they match the journal requirements in terms of DPI.**

Reviewer 2.
Page 6, the authors state that "The OT technique is nominally less affected by temporal signal decorrelation than the InSAR technique. Therefore, we used all possible image couples, leading to the creation of 351 correlograms in both range and azimuth directions. Then, we applied a stacking procedure to create one velocity map in the range direction and one in the azimuth direction." Is the matching identical for all couples? In other words, how is it ensured that the same points are identified for all image couples? And how are the offset values related to the matching point couples combined into one velocity map? A more detailed description of the algorithm should be provided.

**Authors: we thank the reviewer for pointing these issues. In the new version of the manuscript, we have improved this paragraphs according to the suggestions made by the reviewer. In the**

**"Data and processing Steps", we added this new paragraph describing the general approach for the stacking procedure, for both unwrapped InSAR phases and OT correlograms:**

**For the unwrapped phases:**

**The stacking procedure starts from a set of unwrapped interferometric phases along with the time interval in days of the SLC-2 relative to the reference SLC-1. The individual unwrapped interferograms are weighted by the time interval in estimating the rate. The underlying assumption is that atmospheric statistics are stationary for the set of N interferograms. The formula for the estimated phase rate is given by:**

$$phase\ rate = \frac{\sum_{j=1}^{N} \Delta t_j \varphi_j}{\sum_{j=1}^{N} \Delta t_j{}^2} \tag{1}$$

**Where $N$ is the number of interferograms, $\varphi_j$ is one given interferogram, $t$ is the time interval inherent with each interferogram (SLC-2 relative to the reference SLC-1). If the matching is not found (low signal coherence), that particular interferogram value is not used in the stacking procedure. So that the same scatterers are identified on many images, but not all the images are used. It follows that the more the images, the more the stacking redundancy, the more the precision of the results.**

**For the Offset tracking:**

**Then, we applied the stacking procedure described above, to create one velocity map in the range direction and one in the azimuth direction. Instead of the unwrapped InSAR phases, we used the pixel offset values from OT correlograms. As for the InSAR phases, if the matching is not found, that particular correlogram value is not used in the stacking procedure. So that the same ground patterns are identified on many images, but not all the images are used.**

Reviewer 2.
Page 7. "Second, we can see a clear pattern of ground motion in the Hell-Bourg area, approximately ± 0.5 m/yr and spatially consistent with ground observations.". The pattern of the Hell-Bourg area looks also quite complex and stating that it is "± 0.5 m/yr" may induce the reader that values are either +0.5 or -0.5 m/yr. It should be specified that ± 0.5 m/yr is the interval of values.

**Authors: we thank the reviewer for pointing these issues. In the new version of the manuscript, we have improved this paragraphs according to the suggestions made by the reviewer.**

Reviewer 2.
Page 7. "In this study, the InSAR signal was surprisingly coherent with a certain level of noise.". This is an interesting result, but should be justified by showing the InSAR coherence map.
**Authors: we agree. In the new version of the manuscript, we have added a coherence map according to the suggestions made by the reviewer.**

Reviewer 2.
The azimuth and LOS OT velocity maps are shown in Figure 3. These maps have a uniform spatial sampling. However, the image matching techniques usually yield a nonuniform grid. Is this true also in this case? If so, was the spatial grid interpolated?

**Authors: In this study we performed the OT measurements on a regular grid (1024 x 1024 measures, in azimuth and LOS directions), on the co-registered SLCs images (radar geometry), before orthorectification. This yields a uniform spatial sampling, in radar geometry. The interpolation comes during the orthorectification process –in order to keep the regular sampling on the final map.**
**We added this sentence in the new paragraph, page 6.**

Reviewer 2.
The notation in equation (1) is not very clear and should be improved using symbols. This applies also to Figure 4.
**Authors: we modified the paragraph as well as figure 4. Now it reads :**

**Therefore, we applied :**

$$|v_h| = \sqrt{OT_{AZ} + (OT_{LOS}/sini)^2},$$
$\qquad$ **(2)**

**to extract the horizontal velocity map, $|v_h|$, where i is the satellite viewing angle ($37°$ in this case study), $OT_{AZ}$ is the azimuth OT and $OT_{LOS}$ is the range OT (Figure 4).**

---

## Author Response (AR2)

BRGM, French Geological Survey,

3 Av. Claude Guillemin, Orléans, France

06 March 2023

Dear Sir Editor of ESURF,

we thank you for the final review of our manuscript and the time you dedicated to evaluate our work. We improved the new version of the manuscript based on your suggestions. We hope that this new, improved, version of the manuscript is suitable for publication in ESURF.

We provide here a point-by-point reply to the Editors suggestions. Our replies are in bold typography; the suggestions made by the Editor are in italic typography.

On behalf of the co-authors,

Marcello de Michele
* * *
BRGM, French Geological Survey - DRP / IGT, 3 avenue Claude Guillemin - 45060 Orléans, France

Direction des Risques et de la Prévention – Imagerie Geophysique et Télédétection - Tel +33.02.38.64.37.95

http://www.researchgate.net/profile/Marcello_De_Michele
* * *
*Editor: The manuscript has been adequately amended in terms of scientific scrutiny. The manuscript is almost ready for publication. I kindly request the authors to now polish the form of the manuscript. Those are very minor requests that are nonetheless important for the credibility and readability of the paper.*

*1- adopt one tense throughout the manuscript. Changes from past to present tenses are a bit confusing at places.*

**Authors: thanks you. We adopted one tense throughout the manuscript, where suitable.**

*2- avoid conjectures in the result section. Move these points of interpretation to the discussion.*

**Authors: we agree. We think the Editor is pointing at the two sentences below, in particular. They were formerly in the result section:**

**"It might correspond to an area that was considered stabilized or dormant, called Crete de Salaze.**

**This area, considered stabilized or dormant, must be investigated further, the OT signal may be due to a post cyclonic burst of ground motion."**

**We moved these sentences in the discussion section of the new version of the manuscript.**